# N-Acetyl-L-cysteine Affects Ototoxicity Evoked by Amikacin and Furosemide Either Alone or in Combination in a Mouse Model of Hearing Threshold Decrease

**DOI:** 10.3390/ijms24087596

**Published:** 2023-04-20

**Authors:** Marek Zadrożniak, Marcin Szymański, Jarogniew J. Łuszczki

**Affiliations:** 1Department of Otolaryngology and Laryngological Oncology, Medical University of Lublin, 20-090 Lublin, Poland; 2Department of Occupational Medicine, Medical University of Lublin, 20-090 Lublin, Poland

**Keywords:** N-acetyl-cysteine, furosemide, amikacin, ototoxicity, hearing loss, hearing threshold decrease

## Abstract

Drug-induced ototoxicity resulting from therapy with aminoglycoside antibiotics and loop diuretics is one of the main well-known causes of hearing loss in patients. Unfortunately, no specific protection and prevention from hearing loss are recommended for these patients. This study aimed at evaluating the ototoxic effects produced by mixtures of amikacin (AMI, an aminoglycoside antibiotic) and furosemide (FUR, a loop diuretic) in the mouse model as the hearing threshold decreased by 20% and 50% using auditory brainstem responses (ABRs). Ototoxicity was produced by the combinations of a constant dose of AMI (500 mg/kg; i.p.) on FUR-induced hearing threshold decreases, and a fixed dose of FUR (30 mg/kg; i.p.) on AMI-induced hearing threshold decreases, which were determined in two sets of experiments. Additionally, the effects of N-acetyl-L-cysteine (NAC; 500 mg/kg; i.p.) on the hearing threshold decrease of 20% and 50% were determined by means of an isobolographic transformation of interactions to detect the otoprotective action of NAC in mice. The results indicate that the influence of a constant dose of AMI on FUR-induced hearing threshold decreases was more ototoxic in experimental mice than a fixed dose of FUR on AMI-induced ototoxicity. Moreover, NAC reversed the AMI-induced, but not FUR-induced, hearing threshold decreases in this mouse model of hearing loss. NAC could be considered an otoprotectant in the prevention of hearing loss in patients receiving AMI alone and in combination with FUR.

## 1. Introduction

Ototoxicity as a result of side effects produced by some specific drugs has been known in medicine for many years. Aminoglycoside antibiotics and loop diuretics are the drugs whose activity is closely correlated with the highest incidence of ototoxic effects in humans. Of note, aminoglycoside antibiotics produce hearing loss and vestibular dysfunction due to permanent hair cell loss [1,2], whereas loop diuretics usually cause hearing loss by acting on the stria vascularis, producing edema and a temporary loss of function, resulting in a decrease in endo-cochlear potential [3]. The molecular processes involved in hair cell survival, death and regeneration as a result of the action of ototoxic agents have been evaluated in various in vivo and in vitro studies [4,5,6,7,8,9].

Many experiments have confirmed that loop diuretics potentiate ototoxic effects in aminoglycoside antibiotics. However, reports are scarce to document the protection from such combinations of drugs. In clinical practice, the co-administration of aminoglycosides (i.e., amikacin [AMI]) along with loop diuretics (i.e., furosemide [FUR]) is usually prescribed for patients undergoing continuous ambulatory peritoneal dialysis (CAPD) with peritonitis [10,11]. In such a situation, ototoxicity, as the well-known complication of such a combined treatment, forced clinicians to prevent the development of ototoxicity. 

Oxidative stress associated with aminoglycoside antibiotics treatment has been extensively investigated in both animal and clinical studies [8,12,13,14]. This commonly used strategy in combat against aminoglycoside ototoxicity is based on the application of antioxidants. Unfortunately, no recommended therapy has been elaborated as yet for hearing loss prevention [15,16], especially if aminoglycosides are combined with loop diuretics.

Overwhelming evidence indicates that N-acetyl-L-cysteine (NAC), due to its anti-inflammatory effects and free radical scavenging properties, exerted protective effects against kanamycin- [17], gentamycin- [18,19,20], and cisplatin-induced ototoxic effects in various animal models [7,21,22,23,24,25,26]. NAC also ameliorated aminoglycoside-induced ototoxicity in dialysis patients [10,27]. The influence of NAC on human health, cell functioning, and the molecular mechanisms of action of NAC has been extensively described elsewhere [28,29,30,31].

Considering molecular mechanisms involved in the development of ototoxicity after treatment with aminoglycosides and loop diuretics, it was crucial to determine the ototoxic effects produced by the mixture of AMI and FUR in an animal model of ototoxicity—the hearing threshold decrease model in mice using auditory brainstem responses (ABRs).

To determine the effects of AMI and FUR on hearing threshold decreases, two sets of experiments were conducted on animals to confirm whether or not any difference in ototoxicity existed in relation to the order and application of drugs (AMI + FUR or FUR + AMI). The second aim of this study was to determine whether NAC could reverse or partially alleviate ototoxicity in the hearing threshold decrease model in mice. The application of NAC was important because of the prophylaxis that could be used before the administration of the drugs with ototoxic side effects.

## 2. Results

### 2.1. Influence of FUR, NAC and Their Combination on AMI-Induced Hearing Threshold Decrease in Mice

AMI administered systemically (i.p.) reduced in a dose-dependent manner the hearing threshold in experimental animals (Figure 1A,C), which allowed for the doses of AMI to be calculated when reducing the hearing threshold in mice by 20% and 50% (i.e., hearing threshold decreasing dose of 20%—HTDD_20_ and hearing threshold decreasing dose of 50%—HTDD_50_) (Table 1).

The statistical analysis of doses for the AMI-induced hearing threshold decrease of 20% (HTDD_20_) and 50% (HTDD_50_) with one-way ANOVA followed by Holm–Sidak’s multiple comparison tests, which revealed that the doses of AMI significantly differed among the tested groups [F(3;20) = 16.47; *p* < 0.0001 for 20%] and [F(3;20) = 31.65; *p* < 0.0001 for 50%], respectively. More specifically, NAC administered i.p. in a constant dose of 500 mg/kg, which shifted (to the right) the dose–response relationship curves for the AMI-induced hearing threshold decrease of 20% and 50%, respectively (Figure 1A,C). In this case, NAC (500 mg/kg) significantly diminished the hearing threshold decrease by 20% (*p* < 0.05) and by 50% (*p* < 0.01) compared to AMI+VEH treated animals (Figure 1B,D). On the other hand, FUR administered i.p. in a constant dose of 30 mg/kg, which shifted to the left the dose–response curve for the AMI-induced hearing threshold decrease (Figure 1A,C), and a statistical analysis of the data revealed that FUR (30 mg/kg) significantly enhanced (*p* < 0.05) the threshold hearing decreases in experimental animals (*p* < 0.001 for 20% and *p* < 0.001 for 50%). In contrast, NAC (500 mg/kg) was added to the mixture of FUR (30 mg/kg) while AMI partially reversed the effects produced by FUR; however, the threshold hearing decrease of 20% for the three-drug combination (AMI+FUR+NAC) did not differ from that for AMI+VEH treated animals (Figure 1B).

Similarly, when NAC (500 mg/kg) was added to the mixture of FUR (30 mg/kg) with AMI for the threshold hearing decrease by 50%, it did not reverse the FUR+AMI-evoked hearing threshold decrease, but the hearing threshold for the three-drug combination of AMI+FUR+NAC significantly differed (*p* < 0.001) from that of AMI+VEH treated animals (Figure 1D). A comparison of the hearing threshold decrease by 20% and 50% for AMI+NAC-treated animals with those for the three-drug combination (i.e., AMI+FUR+NAC) revealed that the addition of NAC partially reversed the hearing threshold decrease of 20% (*p* < 0.01; Figure 1B), and simultaneously, NAC did not reverse that which was reported for the hearing threshold decrease of 50% (*p* < 0.0001; Figure 1D). In this case, the effects produced by the addition of FUR to AMI evoked a hearing threshold decrease that was not alleviated by NAC.

### 2.2. Influence of AMI, NAC and Their Combination on FUR-Induced Hearing Threshold Decrease in Mice

FUR administered i.p., which diminished, in a dose-dependent manner, the hearing threshold in experimental animals (Figure 2A,C), which allowed the doses of FUR to be calculated, which reduced the hearing threshold in mice by 20% and 50% (i.e., HTDD_20_ and HTDD_50_) (Table 2).

Statistical analysis of doses for the FUR-induced hearing threshold decrease of 20% (HTDD_20_) with one-way ANOVA followed by Holm–Sidak’s multiple comparison tests, which revealed that doses of FUR significantly differed among the tested groups [F(3;20) = 9.729; *p* = 0.0004]. Similarly, one-way ANOVA with Holm–Sidak’s post hoc test revealed that doses of the FUR-induced hearing threshold decrease of 50% (HTDD_50_) and also considerably differed [F(3;20) = 7.206; *p* = 0.0018]. NAC administered i.p. in a constant dose of 500 mg/kg, which partially protected the mice from hearing threshold decreases compared to AMI+VEH treated animals, but this effect was not statistically significant for both a 20% and 50% hearing threshold decrease (Figure 2B,D). In contrast, AMI administered i.p. in a constant dose of 500 mg/kg significantly potentiated (*p* < 0.05) the threshold hearing decreases in experimental animals for a hearing threshold decrease of 20%, but not that for a hearing threshold decrease of 50%. In the case of NAC (500 mg/kg), when added to the mixture of AMI (500 mg/kg) with FUR, it did not reverse the effects produced by AMI (500 mg/kg) because the threshold hearing decrease for 20% for the three-drug combination (AMI+FUR+NAC) did not differ for the FUR+AMI treated animals (Figure 2B). Simultaneously, the threshold hearing decrease for 20% of the combination (AMI+FUR+NAC) considerably differed from that for FUR+VEH-treated animals (*p* < 0.05; Figure 2B). However, when NAC (500 mg/kg) was added to the mixture of AMI (500 mg/kg) with FUR for the threshold hearing decrease of 50%, it did not reverse the AMI+FUR-evoked hearing threshold deficits, and the hearing threshold for the three-drug combination of FUR+AMI+NAC did not differ from that for FUR+VEH treated animals (Figure 2D). A comparison of the hearing threshold decreases by 20% and 50% for FUR+NAC-treated animals with those for the three-drug combination (i.e., FUR+AMI+NAC) revealed that NAC administration partially reversed the hearing threshold decrease of 20% (*p* < 0.01; Figure 2B) and by 50% (*p* < 0.05; Figure 2D). In this case, the effects produced by the addition of AMI to FUR a evoked hearing threshold decrease that was not alleviated by NAC.

### 2.3. Isobolographic Transformation of Interaction between AMI, FUR and NAC in the Drug-Induced Hearing Threshold Decrease Model in Mice

AMI in a constant dose of 500 mg/kg when combined with increasing doses of FUR diminished in a dose-dependent manner the hearing threshold in experimental animals, and the experimentally derived HTDD_20_ and HTDD_50_ values for the mixture of both ototoxic drugs were placed into the Cartesian coordination system to illustrate the interactions between these two drugs (point B) and NAC, which was additionally added to the mixture of both ototoxic drugs (point C) (Figure 3a,b). NAC evidently elevated the doses of FUR that were required to produce the hearing threshold decrease of 20% and 50% in tested animals (Figure 3a,b). With Student’s *t*-test, it was found that the difference between both FUR values (points B and C) was significant at *p* < 0.001 (t = 5.689; df = 10; *p* = 0.0002) for a hearing threshold decreased by 20% (Figure 3a) and at *p* < 0.01 (t = 4.264; df = 10; *p* = 0.0017) for a hearing threshold decreased by 50% (Figure 3b). The isobolographic transformation of interaction for AMI and FUR revealed that the experimentally derived HTDD_20_ and HTDD_50_ for AMI+FUR (points B) considerably differed from the theoretically calculated additive doses, reducing the hearing threshold in animals (points A). More specifically, FUR was administered in higher (statistically significant) doses than theoretically expected to produce a hearing threshold decrease of 20% and 50% when combined with a constant dose of AMI (500 mg/kg; Figure 3a,b). With Student’s *t*-test, it was found that the difference between both FUR values (points A and B) was significant at *p* < 0.05 (t = 2.675; df = 7.595; *p* = 0.0295) for the hearing threshold decreased by 20% (Figure 3a) and at *p* < 0.05 (t = 2.472; df = 9.392; *p* = 0.0344) and for a hearing threshold decreased by 50% (Figure 3b).

Similarly, FUR in a constant dose of 30 mg/kg, when combined with increasing doses of AMI, reduced in a dose-dependent manner the hearing threshold in experimental animals, and the experimentally derived HTDD_20_ and HTDD_50_ values for the mixture of both ototoxic drugs were placed into the Cartesian coordination system to illustrate the interactions between these two drugs (point B) and NAC, which was added to the mixture of both ototoxic drugs (point C) (Figure 3c,d). NAC evidently elevated the doses of AMI that were required to produce a hearing threshold decrease of 20% and 50% in the tested animals (Figure 3c,d). With Student’s *t*-test, it was found that the difference between both AMI values (points B and C) was significant at *p* < 0.001 (t = 5.360; df = 10; *p* = 0.0003) for a hearing threshold decrease of 20% (Figure 3c) and at *p* < 0.0001 (t = 11.87; df = 10; *p* < 0.0001) for a hearing threshold decrease of 50% (Figure 3d). The isobolographic transformation of the interaction for FUR and AMI revealed that the experimentally derived HTDD_20_ and HTDD_50_ for FUR+AMI (point B) considerably differed from the theoretically calculated doses, reducing the hearing threshold in animals by 20% (point A; Figure 3c) but not in animals with a hearing threshold decrease of 50% (point A; Figure 3d). More specifically, AMI was administered in higher (statistically significant) doses than theoretically expected to produce a hearing threshold decrease of 20% when combined with a constant dose of FUR (30 mg/kg; Figure 3c). In contrast, AMI was administered in similar doses than theoretically expected to produce a hearing threshold decrease of 50% when combined with a constant dose of FUR (30 mg/kg; Figure 3d). With Student’s *t*-test, it was found that the difference between both AMI values (points A and B) was significant at *p* < 0.001 (t = 4.698; df = 16.78; *p* = 0.0002) for a hearing threshold decrease of 20% (Figure 3c) but was not significant (t = 0.304; df = 17.17; *p* = 0.7648) for a hearing threshold decrease of 50% (Figure 3d).

## 3. Discussion

Accumulative evidence indicates that some antioxidants (including NAC, nicotinamide riboside, taxifolin, resveratrol, vitamin C, salvianolic acid B, and apocynin) ameliorated the ototoxic effects evoked by aminoglycosides (kanamycin, neomycin, amikacin, gentamycin) in various experimental models of ototoxicity [17,32,33,34,35,36,37,38,39,40]. For instance, NAC administered chronically for 15 days in a dose of 500 mg/kg, i.p., protected the rats from gentamicin-induced ototoxicity, which was confirmed both acoustically (ABRs monitoring) and histologically (low level of apoptosis in cochlear cells) [19]. NAC administered 24 h prior to gentamycin exposure in rat cochlea explants (in an in vitro study) protected both the outer and inner hair cells against gentamycin-induced ototoxicity [20]. However, in patients, only two clinical trials have been completed, for which the results have not yet been posted, in which clinicians investigated the otoprotective properties of NAC in aminoglycosides-induced ototoxicity (NTC01271088, NTC01131468) [33]. Additionally, a novel clinical trial investigating the otoprotective effects of ORC-13661 in intravenous AMI-induced ototoxicity has been posted but not recruited by the patients yet, (NCT05730283) [41]. Considering the fact that no recommendation was formulated about the protection against aminoglycoside-induced ototoxicity, NAC was tested in both preclinical and clinical trials, and we chose NAC in our isobolographic study to confirm its preventive and prophylactic role in further otoprotective therapy.

In this study, we conducted two sets of experiments related to the order of drug administration. In the first set of experiments, we determined the effects of constant doses of NAC and FUR on AMI-induced ototoxicity, whereas, in the second set of experiments, we determined the effects of NAC and AMI on the FUR-evoked hearing threshold decrease in experimental animals. In these two sets of experiments, we found evidently that the order of administration of ototoxic drugs was very important and crucial for observed ototoxic effects. NAC alleviated the ototoxicity in AMI-treated animals but not in FUR-treated mice. We confirmed that NAC reversed the ototoxicity evoked by AMI but not that of FUR, suggesting that this compound could be used as a prophylaxis for AMI-induced ototoxicity. In this study, we found that FUR evoked a hearing threshold decrease and was resistant to NAC. 

Moreover, we confirmed that hearing threshold changes produced by AMI and FUR (either alone or in combination) were strongly expressed when testing in animals in the model of a hearing threshold decrease of 50% compared to that of 20%. It is important to note that a hearing threshold decrease of 20% was investigated in animals, which received respective treatment 15 min prior to the hearing threshold measurement, whereas the hearing threshold decrease of 50% was detected in the mice receiving the respective treatment 30 min before the hearing threshold measurement. It seems that in this animal model, more time was needed for the drugs to reach their targets in the inner ear in order to diminish the hearing threshold. In both sets of experiments, NAC was injected systemically (i.p.) at 60 min before the hearing threshold detection. It is important to note that this animal model based on the hearing threshold decrease of 20% and 50% was applied in experimental studies to determine the influence of drugs with ototoxic properties on hearing processes and any subtle effects that reduced the hearing threshold by 20% and 50% were investigated. Of note, in this experimental model, each animal was examined twice, the first time before the drug administration (pretreatment threshold), which was considered to be a baseline threshold, and the second examination was conducted after the drug treatment (posttreatment threshold). An experimental evaluation of the hearing threshold confirmed that hearing deficits resulted from drug administration but not from congenital deafness in animals.

Another fact related to the selection of constant doses of FUR and AMI, when combined together, needs a short explanation. Previously, we determined the doses of AMI and FUR, which, when used alone, produced by themselves a hearing threshold decrease of 50% in experimental animals. For the combinational experiments, we selected half of the doses of FUR and AMI, which decreased the hearing threshold in animals by 50%. This was the reason to test the effects produced by FUR in a constant dose of 30 mg/kg and AMI in a constant dose of 500 mg/kg, which were approx. their halves of the doses and decreased the hearing threshold by 50% in animals. We used the same AMI and FUR doses to detect the hearing threshold decrease by 20% in animals due to a simple comparison of the results among the respective groups of animals.

Isobolographic transformation revealed that all the interactions between AMI and FUR were antagonistic in nature, except for one combination of FUR+AMI in the hearing threshold decreased by 50%, which was additive. The antagonistic interaction between FUR and AMI confirmed at least in part that both drugs concurrently antagonized their own effects and, thus, more drugs were required to produce a hearing threshold decrease of 20% and 50% in animals compared to theoretically calculated [42], and presumed to be additive ototoxic doses. In the case of the constant dose of FUR 30 mg/kg, which was added to the AMI, the mixture of both drugs exerted additive effects in the hearing threshold decrease of 50% in mice. FUR in this dose (30 mg/kg) was sufficiently effective to exert additivity in this model. Noteworthy, AMI and FUR exerted their effects not only in the inner ears but also in the kidneys, where FUR inhibited the excretion of AMI, increasing the AMI content not only in the serum but also in perilymph, cerebrospinal fluid and the inner ears [43]. Thus, increased concentrations of AMI could be responsible for the observed interactions between these two drugs in mice. FUR was combined with AMI in the two sets of experiments, but only in one case was additivity observed when FUR was added in a fixed dose of 30 mg/kg, while, in other cases, antagonistic interactions occurred. The most probable explanation for the antagonistic interaction between AMI and FUR or FUR and AMI is that FUR in higher doses than 30 mg/kg reduced the volume of fluids (serum and other body fluids). Dehydration and electrolytic changes in the inner ears reduced the penetration of AMI to the inner ears and, thus, produced oxidative stress and changes in ROS in cochlear tissue. A reduction in blood pressure in response to high doses of FUR slowed down the penetration of AMI to its target site in the inner ears. Although this hypothesis is very speculative, it can readily explain the observed effects produced by the combination of AMI and FUR in animals, as illustrated isobolographically. Additionally, the application of NAC, partially when considering the reversed effects exerted by AMI, changed the hearing threshold decrease in animals.

Generally, NAC is able to reverse the effects produced by AMI in terms of the restoration of oxidative potential in cochlear cells. In such a situation, the hearing threshold decrease of 20% and 50% were shifted to higher doses of FUR or the mixture of AMI with FUR to break the protective effects of NAC. Of note, NAC was administered in a constant dose of 500 mg/kg in both sets of experiments, in which doses of FUR or AMI increased proportionally to the observed hearing threshold decreases of 20% and 50%.

The main limitation of this study was the acute experimental administration of the drugs (AMI and FUR alone and in combination with NAC). In this study, we evaluated the hearing threshold by 20% and 50% after acute (single) administration of the drugs in combination. Acute vs. chronic administration of the ototoxic drugs are very important in relation to the fact that loop diuretics usually evoke reversible hearing loss due to their influence on sodium, chloride and potassium cotransporters in the cochlea and temporal induction of ions imbalance in the endolymphatic fluid of the inner ears [44]. By contrast, aminoglycoside antibiotics, due to their destructive effects on outer hair cells, induce irreversible hearing loss in individuals receiving aminoglycosides [44]. On the other hand, the effects of both groups of drugs when used together were not tested experimentally, and little is known about the effects exerted by the drugs used concomitantly and chronically.

## 4. Materials and Methods

### 4.1. Animals and Experimental Conditions 

Male (14–21 days old) Albino Swiss outbred mice were used in this study. After adaptation to laboratory conditions, 144 mice were randomly assigned to experimental groups containing 6 animals (24 groups). The experimental protocols and procedures described herein were approved by the Local Ethics Committee (License no. 31/2008) and were in accordance with the ARRIVE guidelines [45].

### 4.2. Drugs

Amikacin (AMI—Biodacyna, Polpharma, Ożarów Mazowiecki, Poland), furosemide (FUR—Furosemidum, Polpharma, Ożarów Mazowiecki, Poland), and *N*-acetyl-L-cysteine (NAC, Sigma-Aldrich, Poznań, Poland) were given intraperitoneally (i.p.) as follows: NAC was given 60 min before AMI and FUR administration, while AMI and FUR were administered i.p. either 15 min (for hearing threshold decrease by 20%) or 30 min (for hearing threshold decrease by 50%) before hearing measurement. The hearing threshold in mice was evaluated in fully anesthetized animals with a combination of xylazine (Xylazinum, Biowet Pulawy, Puławy, Poland—in a dose of 20 mg/kg (i.p.)) and ketamine (Ketamini hydrochloridi, Vetoquinol Biowet, Gorzów Wielkopolski, Poland—in a dose of 100 mg/kg (i.p.)).

### 4.3. Auditory Brainstem Responses (ABRs)

Auditory brainstem responses (ABRs) were collected using a computerized Interacoustics Eclipse EP15 evoked potential unit (Middelfart, Denmark). The left ear of each animal was stimulated by alternating clicks, and ABRs were recorded via subcutaneous electrodes placed near the ipsilateral pinna, vertex and contralateral pinna, with the ground electrode placed along the trunk. The responses were amplified, filtered, averaged by a computer, and displayed on a computer screen. All the detailed information about the ABRs and their analysis is described elsewhere [39]. Briefly, the sound level of the stimuli decreased from a 90 dB sound pressure level (SPL) to 20 dB SPL in 10 dB steps and finally in 5 dB steps to identify the lowest intensity at which an ABR wave V was detectable. At each sound level, up to 1000 responses were averaged and analyzed by a computerized unit. If the repeatability of the recorded wave exceeded 95%, the system automatically started the stimulation with the next programmed volume level. Stimuli were presented at a rate of 39 Hz and were recorded for a 10 ms duration. The hearing threshold was determined by a single observer (blinded to the respective treatment), who noted the lowest sound level at which a recognizable waveform was seen on a screen from the highest to the lowest sound levels. Waveforms were confirmed as auditory evoked responses by their decreasing amplitude and increased latency with the decreasing sound intensity of the stimulus and repeatability, which was at least 95%. During the evaluation of the hearing threshold, the anesthetized animals were maintained on a warming pad to keep the animals’ rectal temperature constant.

### 4.4. Calculations of Hearing Threshold

A pretreatment hearing threshold was assessed in each anesthetized mouse before the systemic (i.p.) administration of the drugs either alone or in combination with NAC. Subsequently, the animals, after receiving AMI and FUR (at increasing doses), were subjected to the measurement of posttreatment hearing threshold. Decreases in the hearing threshold (in %) were calculated by comparing the pretreatment with posttreatment hearing thresholds. The percentage of the hearing threshold decrease for the respective doses of a drug mixture (AMI with FUR or FUR with AMI) was linearly assessed, from which the hearing threshold doses decreased by 20% and 50% (HTDD_20_ and HTDD_50_) were calculated for AMI, FUR and their combination. To illustrate dose–response effects between the increasing doses of the studied drugs and their corresponding decreases in the hearing threshold, sigmoidal dose–response curves were constructed, from which the HTDD_20_ and HTDD_50_ values were calculated. To determine the HTDD_20_ and HTDD_50_ values, at least 3 points were plotted, and the curves were best fit to these points, reflecting particular doses of the studied drugs when administered either alone or in combination with their corresponding decreases in the hearing threshold. From a pharmacological viewpoint, to adequately construct each sigmoidal curve, at least 3 drug doses need to be used [46,47]. Of note, 3 points were optimal to determine sigmoidal dose–response effects; however, many more points can be sometimes necessary to construct and best fit the dose–response curve [46,47]. It should be stated that, in this study, drug doses were selected in a such manner that their corresponding decrease in the hearing threshold was lower and higher than the calculated 20% and 50%, respectively. Additionally, neither minimal (0%) nor maximal (100%) hearing threshold effects were used in this study to calculate the HTDD_20_ and HTDD_50_. From 3 points, it was possible to determine the HTDD_20_ and HTDD_50_ values, and this was the reason that only 3 doses of each drug administered alone or in combination with NAC were investigated. Of note, the pretreatment hearing threshold was predicted to be 100%, and any changes in the animals’ hearing threshold after the injection of AMI and FUR were expressed as the % of reduction in the hearing threshold, as described elsewhere [39].

### 4.5. Isobolographic Transformation

The effects exerted by the mixture of AMI and FUR with NAC underwent isobolographic transformation. Any changes in the hearing threshold were calculated using linear regression analysis. The influence of NAC (500 mg/kg, i.p.) on the hearing threshold in experimental mice was assessed by estimating the hearing threshold for animals receiving three drugs (AMI, FUR and NAC), from which the experimentally decreases in the hearing threshold were 20% and 50% (HTDD_20_ and HTDD_50_) and calculated as presented elsewhere [48,49,50].

### 4.6. Statistical Analysis

Experimentally determined hearing threshold decreasing doses of 20% and 50% (HTDD_20_ and HTDD_50_) for AMI and FUR were analyzed with one-way ANOVA followed by post hoc Holm–Sidak’s multiple comparison tests. The statistical comparison of isobolographically transformed additive and experimental values was performed with Student’s *t*-test as presented earlier [51]. Differences among HTDD_20_ and HTDD_50_ values were considered statistically significant if *p* < 0.05.

## 5. Conclusions

NAC reversed the ototoxic effects evoked by AMI in the mouse model of the hearing threshold decrease by 20% and 50%. In contrast, NAC had no impact on the ototoxicity evoked by FUR in experimental animals. Only the isobolographic transformation of interaction should be used in combinatorial studies when evaluating the ototoxic effects produced by drug mixtures, whose side effects contribute to ototoxicity in experimental animals. The evaluation of the hearing threshold decrease by 20% and 50% seemed to be a good experimental model allowing for the detection of any subtle changes in drug-induced ototoxicity in animals.

## Figures and Tables

**Figure 1 ijms-24-07596-f001:**
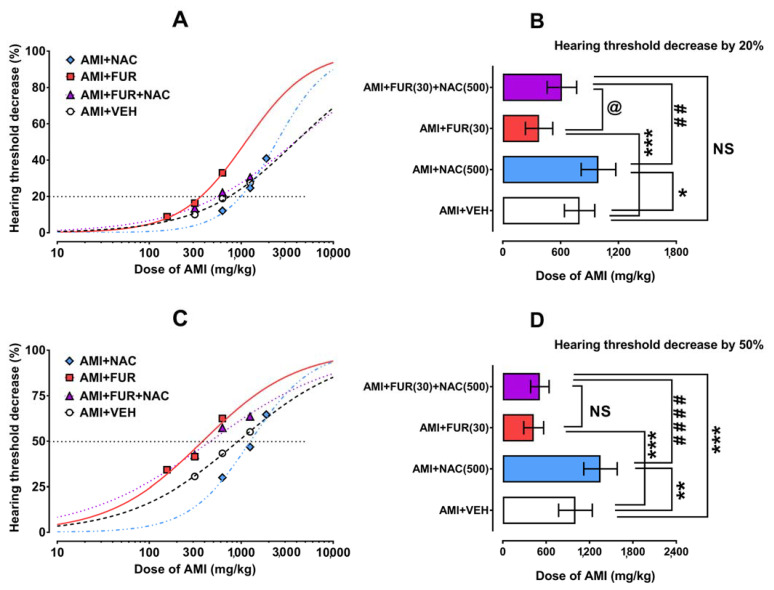
Effects of NAC, FUR and their combinations on AMI-induced hearing threshold decrease in mice. The dose–response relationship curves for AMI-induced hearing threshold decrease of 20% (**A**) and 50% (**C**) are plotted graphically, from which the HTDD_20_ and HTDD_50_ values (for AMI administered alone and in combination with FUR, NAC, and both compounds) were calculated. The HTDD_20_ (**B**) and HTDD_50_ (**D**) values were statistically analyzed with one-way ANOVA followed by Holm–Sidak’s multiple comparisons tests. * *p* < 0.05, ** *p* < 0.01 and *** *p* < 0.001 vs. AMI+VEH-treated animals; ## *p* < 0.01 and #### *p* < 0.0001 vs. AMI+NAC-treated animals; @ *p* < 0.05 vs. AMI+FUR-treated animals; NAC—N-acetylcysteine; AMI—amikacin; FUR—furosemide; NS—nonsignificant; VEH—vehicle.

**Figure 2 ijms-24-07596-f002:**
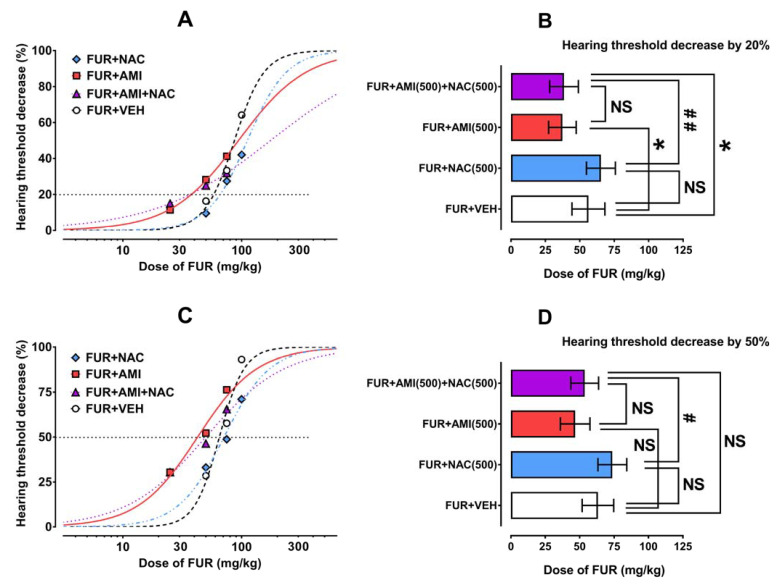
Effects of NAC, AMI and their combinations on FUR-induced hearing threshold decrease in mice. Dose–response relationship curves for FUR-induced hearing threshold decrease of 20% (**A**) and 50% (**C**) are plotted graphically, from which the HTDD_20_ and HTDD_50_ values (for FUR administered alone and in combination with AMI, NAC, and both compounds) were calculated. The HTDD_20_ (**B**) and HTDD_50_ (**D**) values were statistically analyzed with one-way ANOVA followed by Holm–Sidak’s multiple comparison tests. * *p* < 0.05 vs. FUR+VEH-treated animals; # *p* < 0.05 and ## *p* < 0.01 vs. FUR+NAC-treated animals. NAC—N-acetylcysteine; AMI—amikacin; FUR—furosemide; NS—nonsignificant; VEH—vehicle.

**Figure 3 ijms-24-07596-f003:**
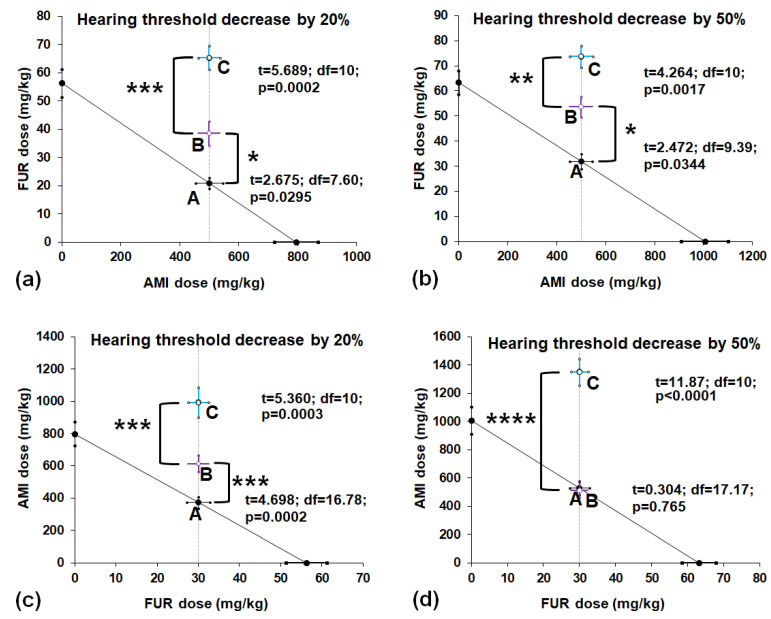
Isobolographic transformation of interaction between AMI and FUR in the drug-induced hearing threshold decrease of 20% and 50% in mice. Doses of AMI and FUR are plotted on the X- and Y-axes, respectively. The dotted line parallel to the *Y*-axis represents a constant dose of AMI (**a**,**b**) or FUR (**c**,**d**), which was added to the increasing doses of FUR or AMI in the mouse model of a hearing threshold decrease of 20% and 50%, respectively. Point A (on each graph) illustrates the theoretically additive dose of the two-drug mixture (AMI+FUR) that could produce a hearing threshold decrease of 20% or 50%, respectively. Point B indicates the dose of AMI and FUR that experimentally evoked a hearing threshold decrease of 20% and 50%, respectively. Point C illustrates the dose of the two-drug mixture producing a hearing threshold decrease of 20% and 50% in animals exposed additionally to NAC (500 mg/kg). * *p* < 0.05, ** *p* < 0.01, *** *p* < 0.001 and **** *p* < 0.0001 vs. the respective treatment group. NAC—N-acetylcysteine; AMI—amikacin; FUR—furosemide.

**Table 1 ijms-24-07596-t001:** Influence of AMI, FUR and NAC on the reduction in hearing threshold in animals receiving the drugs either alone or in combination.

Treatment	HTDD_20_	HTDD_50_
AMI+VEH	797 ± 64	1006 ± 95
AMI+NAC(500)	993 ± 73 *	1352 ± 94 **
AMI+FUR(30)	377 ± 58 ***, ####	428 ± 56 ***, ####
AMI+FUR(30)+NAC(500)	614 ± 62 ##, @	513 ± 52 ***, ####
	F(3;20) = 16.47; *p* < 0.0001	F(3;20) = 31.65; *p* < 0.0001

Results are presented as doses (in mg/kg) of AMI, which decreased the hearing threshold in mice by 20% and 50% (HTDD_20_ and HTDD_50_). A reduction in the hearing threshold was calculated by comparing the hearing threshold values denoted experimentally for the combination of AMI, FUR and NAC (posttreatment hearing threshold) with those determined in the same animals prior to the administration of the drugs (at the pretreatment hearing threshold). NAC (500 mg/kg, i.p.) was added to AMI, FUR or their combination (AMI+FUR). AMI—amikacin; FUR—furosemide; NAC—N-acetylcysteine; VEH—vehicle. * *p* < 0.05, ** *p* < 0.01 and *** *p* < 0.001 vs. AMI+VEH-treated animals; ## *p* < 0.01 and #### *p* < 0.0001 vs. AMI+NAC-treated animals; @ *p* < 0.05 vs. AMI+FUR-treated animals.

**Table 2 ijms-24-07596-t002:** Influence of FUR, AMI and NAC on the reduction in hearing threshold in animals receiving the drugs either alone or in combination.

Treatment	HTDD_20_	HTDD_50_
FUR+VEH	56.25 ± 4.89	63.24 ± 4.68
FUR+NAC(500)	65.31 ± 4.29	73.72 ± 4.28
FUR+AMI(500)	37.35 ± 4.10 *, ##	46.73 ± 4.39 ##
FUR+AMI(500)+NAC(500)	38.52 ± 4.26 *, ##	53.64 ± 4.13 #
	F(3;20) = 9.729; *p* = 0.0004	F(3;20) = 7.206; *p* = 0.0018

The results are presented as doses (in mg/kg) of FUR, which decreased the hearing threshold in mice by 20% and 50% (HTDD_20_ and HTDD_50_). The reduction in the hearing threshold was calculated by comparing the hearing threshold values denoted experimentally for the combination of FUR, AMI and NAC (posttreatment hearing threshold) with those determined in the same animals prior to the administration of the drugs (pretreatment hearing threshold). NAC (500 mg/kg, i.p.) was added to AMI, FUR or their combination (FUR+AMI). AMI—amikacin; FUR—furosemide; NAC—N-acetylcysteine; VEH—vehicle. * *p* < 0.05 vs. FUR+VEH-treated animals; # *p* < 0.05 and ## *p* < 0.01 vs. FUR+NAC-treated animals.

## Data Availability

Data are contained within the article.

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
