# Peer review of "N-Acetyl-L-cysteine Affects Ototoxicity Evoked by Amikacin and Furosemide Either Alone or in Combination in a Mouse Model of Hearing Threshold Decrease"

_ijms, 2023, doi:10.3390/ijms24087596_

Round 1

Reviewer 1 Report

Zadrozniak et al examined the ototoxicity induced by a mixture of amikacin (AMI) and furosemide (FUR), as well as its prevention with N-acetylcysteine (NAC), using an auditory brainstem response (ABR) test in a mouse model. Based on the results of the evaluation of hearing threshold decreases by 20% and 50%, as well as its isobolographic analysis, the authors suggested that NAC exerts preventive effects against AMI-induced ototoxicity, but has little effects on FUR-induced ototoxicity. The concept of this pharmacological study is important for researchers in this field. However, the following concerns should be addressed before publication.

1. The reviewer suggests that the authors revise the discussion section to be more concise, as information from the results section is frequently repeated. Additionally, the discussion should clearly state the new findings in comparison to the authors' previous work (PMID: 30831441).

2. The reviewer suggests that the detailed information on how to illustrate the sigmoid curves shown in Figures 1 and 2 should be included in the Methods and Materials section. Each curve appears to have been calculated based on three points of data, but the authors should clearly explain whether this number of points is sufficient or not. The information is quite critical in this pharmacological study.

3. The authors should consider whether the "ZOOM in" illustrated on the right-hand side of Figures 3a-d respectively is necessary in the revised manuscript.

Author Response

Comments and Suggestions for Authors

Zadrozniak et al examined the ototoxicity induced by a mixture of amikacin (AMI) and furosemide (FUR), as well as its prevention with N-acetylcysteine (NAC), using an auditory brainstem response (ABR) test in a mouse model. Based on the results of the evaluation of hearing threshold decreases by 20% and 50%, as well as its isobolographic analysis, the authors suggested that NAC exerts preventive effects against AMI-induced ototoxicity, but has little effects on FUR-induced ototoxicity. The concept of this pharmacological study is important for researchers in this field. However, the following concerns should be addressed before publication.

  1. The reviewer suggests that the authors revise the discussion section to be more concise, as information from the results section is frequently repeated. Additionally, the discussion should clearly state the new findings in comparison to the authors' previous work (PMID: 30831441).

Reply:

Page 7: We have deleted the repeated information from the results section as requested.

  1. The reviewer suggests that the detailed information on how to illustrate the sigmoid curves shown in Figures 1 and 2 should be included in the Methods and Materials section. Each curve appears to have been calculated based on three points of data, but the authors should clearly explain whether this number of points is sufficient or not. The information is quite critical in this pharmacological study.

Reply:

Pages 10-11: To illustrate dose-response effects between the increasing doses of the studied drugs and their corresponding decreases in the hearing threshold, sigmoidal dose-response curves were constructed, from which the HTDD20 and HTDD50 values were calculated. To determine the HTDD20 and HTDD50 values, at least 3 points were plotted and the curves were best-fit to these points, reflecting particular doses of the studied drugs when administered either alone or in combinations with their corresponding decreases in hearing threshold. From a pharmacological viewpoint, to adequately construct each sigmoidal curve, at least 3 drug doses should be used. Of note, 3 points are optimal to determine 20% and 50% effects, however, many more points can be sometimes necessary to construct and best-fit the dose-response curve. It should be stated that in this study, the drug doses were selected in a such manner so as to their corresponding decreases in the hearing threshold were lower and higher than the calculated 20% and 50%. Additionally, neither minimal (0%), nor maximal (100%) hearing threshold effects were accepted in this study. From 3 points, it was possible to determine the HTDD20 and HTDD50 values and this is the reason that only 3 doses of each drug administered alone or in combination with NAC were investigated. Of note, experiments on animals must follow the ARRVIVE guidelines and the 3R’s rule (Replacement, Reduction and Refinement). Taking together the mentioned facts, 3 doses were chosen as an experimental compromise between the researcher’s 3R’s obligation to reduce number of animals used in this study and the pharmacological methodology to best-fit the results when creating the particular dose-response curves. This is the main reason to test only 3 doses of drugs along with their hearing threshold decreases (3 points) and measure the ABRs from which HTDD20 and HTDD50 values were determined.   

  1. The authors should consider whether the "ZOOM in" illustrated on the right-hand side of Figures 3a-d respectively is necessary in the revised manuscript.

Reply:

We have removed the “zoom in” figures from the multipart Figure 3, as requested.

Reviewer 2 Report

The article is well written, the Authors present an interesting study. Congratulations to the Authors.

Recommendations:

  1. Move the Materials and Methods section after the Introduction section.
  2. In the Introduction section, you should further explain the mechanism of action of Acetylcysteine. Furthermore, you should briefly summarize the routine use of Acetylcysteine in the medical field.

Author Response

Recommendations:

  1. Move the Materials and Methods section after the Introduction section.

Reply:

When the manuscript was prepared we have used an original template.

  1. In the Introduction section, you should further explain the mechanism of action of Acetylcysteine. Furthermore, you should briefly summarize the routine use of Acetylcysteine in the medical field.

Reply:

Page 2: Since molecular mechanisms of action of NAC are widely discussed by other authors in various publications, we have inserted some specific citations, describing in details molecular mechanisms of action of the drug.